# Comparative Yield of Tuberculosis during Active Case Finding Using GeneXpert or Smear Microscopy for Diagnostic Testing in Nepal: A Cross-Sectional Study

**DOI:** 10.3390/tropicalmed6020050

**Published:** 2021-04-14

**Authors:** Suman Chandra Gurung, Kritika Dixit, Bhola Rai, Raghu Dhital, Puskar Raj Paudel, Shraddha Acharya, Gangaram Budhathoki, Deepak Malla, Jens W. Levy, Knut Lönnroth, Andrew Ramsay, Buddha Basnyat, Anil Thapa, Gokul Mishra, Bishal Subedi, Mohammad Kashim Shah, Anil Shrestha, Maxine Caws

**Affiliations:** 1Birat Nepal Medical Trust, Kathmandu 44600, Nepal; scgurung@bnmt.org.np (S.C.G.); bhola@bnmt.org.np (B.R.); raghu.dhital@bnmt.org.np (R.D.); puskar@bnmt.org.np (P.R.P.); shraddha@bnmt.org.np (S.A.); ganga@bnmt.org.np (G.B.); deepak@bnmt.org.np (D.M.); gokulmishra@gmail.com (G.M.); maxine.caws@lstmed.ac.uk (M.C.); 2LIV-TB Collaboration, Departments of International Public Health and Clinical Sciences, Liverpool School of Tropical Medicine, Liverpool L35QA, UK; 3Department of Global Public Health, WHO Collaborating Center on Tuberculosis and Social Medicine, Karolinska Institutet, 10653 Stockholm, Sweden; knut.lonnroth@ki.se; 4KNCV Tuberculosis Foundation, 2514 JD The Hague, The Netherlands; jens.levy@kncvtbc.org; 5School of Medicine, University of St Andrews, St Andrews KY16 9TF, UK; andy.ramsay@st-andrews.ac.uk; 6Oxford University Clinical Research Unit, Kathmandu 44600, Nepal; buddhabasnyat@gmail.com; 7National Tuberculosis Centre, Kathmandu 44600, Nepal; anilthp@gmail.com; 8Health Office, Pyuthan 22300, Nepal; bishal.subedi8@gmail.com; 9Nick Simons Institute, Lalitpur 44600, Nepal; kashimshah@gmail.com (M.K.S.); anilsh@nsi.edu.np (A.S.)

**Keywords:** active case finding, TB yield rate, ACF strategies, ACF interventions, GeneXpert, Nepal

## Abstract

This study compared the yield of tuberculosis (TB) active case finding (ACF) interventions applied under TB REACH funding. Between June 2017 to November 2018, Birat Nepal Medical Trust identified presumptive cases using simple verbal screening from three interventions: door-to-door screening of social contacts of known index cases, TB camps in remote areas, and screening for hospital out-patient department (OPD) attendees. Symptomatic individuals were then tested using smear microscopy or GeneXpert MTB/RIF as first diagnostic test. Yield rates were compared for each intervention and diagnostic method. We evaluated additional cases notified from ACF interventions by comparing case notifications of the intervention and control districts using standard TB REACH methodology. The project identified 1092 TB cases. The highest yield was obtained from OPD screening at hospitals (n = 566/1092; 52%). The proportion of positive tests using GeneXpert (5.5%, n = 859/15,637) was significantly higher than from microscopy testing 2% (n = 120/6309). (OR = 1.4; 95%CI = 1.12–1.72; *p* = 0.0026). The project achieved 29% additionality in case notifications in the intervention districts demonstrating that GeneXpert achieved substantially higher case-finding yields. Therefore, to increase national case notification for TB, Nepal should integrate OPD screening using GeneXpert testing in every district hospital and scale up of community-based ACF of TB patient contacts nationally.

## 1. Introduction

Tuberculosis (TB) is an ancient disease which remains one of the most intractable public health challenges. In 2019, approximately 10 million people developed TB, and 1.5 million died from this preventable, curable disease [1]. One of the reasons for the persistence of the TB pandemic is our collective failure to diagnose and effectively treat all cases. In 2019, over three million cases remained missing from government National TB Programme (NTP) notifications [1]. To accelerate progress towards the END-TB strategy target of reducing TB incidence by 90% by 2035, WHO has strongly recommended high-incidence countries scale up active case finding (ACF) and systematic screening in specific risk populations such as close contacts of index patients [2,3]. However, integrating ACF into existing healthcare services is important to optimize resources and achieve sustainability [4].

WHO recommends a rapid molecular diagnostic such as GeneXpert MTB/RIF (Cepheid, Sunnyvale, CA, USA) as the first diagnostic test in any presumptive TB case [5]. However, the relatively high cost of GeneXpert compared to traditional smear microscopy diagnosis has limited uptake by many low-income countries. The test is often only applied to groups at high risk for multidrug-resistant TB (MDR-TB) and HIV positive cases. Nevertheless, if routinely applied, accurate rapid diagnostic tests have great potential for increasing case identification and early treatment. We, therefore, sought to compare the difference in case-finding yields of ACF strategies applying either traditional smear microscopy or GeneXpert testing, which is the primary diagnostic test in Nepal. We also compared the yield of three different case-finding interventions in Nepal: (1) district hospital OPD screening, (2) community-based social contact tracing of known index TB patient, and (3) TB camps in remote or underserved populations.

Nepal is a Himalayan country in South Asia with a vibrant diverse culture and a significant burden of TB (annual incidence 245/100,000) [6]. The NTP provides routine free TB diagnosis and treatment services using the passive case-finding (PCF) model with sporadic ACF activities implemented under varying models and in different geographical areas, by the NTP and partner organisations [7]. Difficult geographical terrain, inadequate knowledge on TB symptoms and services, and poverty have limited access to TB services among the general population, particularly in remote areas [8,9]. The 2019 Nepal prevalence survey estimated that 40,000 cases are missing from the NTP notifications each year [6]. Therefore, the NTP strategy 2016-21 has prioritized ACF scale-up as a diagnostic strategy to complement PCF [10]. The strategy recommended conducting screening among close contacts of index patients, TB camps, and hospital screening. Nevertheless, there remains a paucity of data on the yield of these strategies and inadequate evidence to drive comprehensive scale up within the NTP, with many competing priorities for government funding. 

Therefore, this study aimed to compare the yield of bacteriologically confirmed TB using either smear microscopy or GeneXpert as the primary diagnostic test and to further compare the yields of the NTP recommended three case-finding strategies. The study also evaluated the additionality achieved from ACF interventions in the study districts. 

## 2. Materials and Methods

Case finding was implemented by Birat Nepal medical trust (BNMT) in eight districts from June 2017 to November 2018 supported by TB REACH wave 5 funding. The districts were purposefully selected from Lumbini province, Karnali province, and Sudurpaschim province, comprising a total population of 2,546,257 (Appendix A).

BNMT installed one four-module GeneXpert machine for TB diagnosis at each of the district government hospitals in Pyuthan, Bardiya, Kapilvastu, and Gulmi districts. The other four districts: Arghakhanchi, Salyan, Doti, and Achham used the standard NTP smear microscopy (Zhiel-Neelsen) for TB diagnosis, which is the primary diagnostic method in Nepal. 

The chief laboratory officer at NTCC provided the hospital laboratory staff in GeneXpert districts with a three day training on the operation GeneXpert machines and in smear districts a 2 day refresher training on standard operating procedures for smear microscopy and quality control of sputum samples. Standard operating procedures for the smear microscopy and GeneXpert testing followed the standard Nepal NTP guidelines, which are based on the relevant WHO protocols and the manufacturer’s standard operating procedures [11,12].

Individuals diagnosed with TB were registered and treated under the routine NTP directly observed treatment shortcourse (DOTS) services. If contacts were negative for TB, they were counselled by the Community Health Volunteers (CHV) and were asked to contact community health volunteers if they developed any symptoms subsequently.

### 2.1. Case-Finding Strategies

The project employed three case-finding strategies: (a) close contact tracing, (b) TB camps in hard to reach areas, and (c) screening at hospital OPD visits (in districts using GeneXpert). All strategies applied the same initial verbal symptom screening using eight questions. Thirty CHVs in each district were trained in the three strategies, including symptom identification and treatment monitoring. The CHVs collected one morning voluntary sputum sample for GeneXpert testing or two samples (spot and morning) for smear microscopy. No sputum induction techniques were used, but participants were given instructions by CHVs on how to produce quality sputum. On the day of sample collection, the CHVs transported the samples to the nearest microscopy or GeneXpert testing centre.

#### 2.1.1. Close-Contact Tracing

CHVs contacted all index cases registered through the NTP in the preceding 12 months and interviewed them to identify name and address of their close contacts. With consent from the index case, the CHV then visited the contacts to screen for cardinal TB symptoms such as cough for more than two weeks, fever, unknown weight loss, night sweats, and lack of appetite [13]. The volunteers collected the morning and on-spot sputum samples of individuals having at least one of these cardinal symptoms and transported the samples to the nearest testing facility.

#### 2.1.2. TB Camps

TB camps were conducted in areas with poor access to TB services, selected in consultation with the government TB officer of the district. The areas were selected based on the high-risk populations such as people from disadvantaged communities, ethnic minorities, and poor socioeconomic conditions. Awareness raising campaigns on TB signs and symptoms were conducted prior to the camps through house-to-house visit or mass media such as newspapers and radio. Local leaders and TB survivors also acted as champions to disseminate information on TB, the planned camp services, and testing to encourage at risk people to attend the camp. The same questionnaire survey was applied to screen camp attendees for TB symptoms. CHV collected sputum samples from symptomatic individuals which were tested by either by GeneXpert or smear microscopy, according to the district.

#### 2.1.3. Screening at Hospital OPD

Verbal symptomatic screening at the hospital OPDs were performed by medical doctors only at the four district hospitals where GeneXpert was installed. Patients who visited the OPD of the district hospital in the four GeneXpert districts for any kind of consultation or diagnosis were screened for symptoms using the eight questions and if symptomatic, were tested for TB.

Five districts Bajhang, Bajura, Banke, Nawalparasi, and Palpa that had similar features such as population size, geographic characteristics, and TB case notification rates were selected as control districts by the external TB REACH monitoring consultant (Appendix A). In these districts, routine passive case finding was implemented following the standard NTP protocols, supplemented by limited household contact tracing supported by the Global Fund [14]. 

##### Ethical Approval

Ethical approvals for the study were received from Liverpool School of Tropical Medicine (LSTM) [N 17-019] and Nepal Health Research Council [149/2017]. Written informed consent was received from participants. Data were anonymised for the analyses.

### 2.2. Statistical Analysis

#### 2.2.1. Screening and Diagnosis of TB Cases

Diagnosed cases were crossverified from the laboratory registers. The primary outcome of the analysis was the yield rate from the three strategies, which was calculated as the number of cases diagnosed divided by the total number of individuals tested. To further evaluate the strategies, the study also compared the number of contacts required to screen (NNS) and the number of contacts required to test (NNT) to identify a TB case from each strategy.

#### 2.2.2. Comparison of Yield from GeneXpert vs. Smear Microscopy

We excluded Salyan and Arghakhanchi districts from the comparative study of yield between GeneXpert and smear microscopy. During the project implementation, the National TB Control Center (NTCC) provided a GeneXpert machine to Salyan district hospital for diagnosis. Similarly, the project discontinued Arghakhanchi district one year after the implementation because of continued delays in project initiation. 

The yield rate from smear microscopy and GeneXpert were compared for close-contact tracing and TB camps. Hospital OPD-based screening was not included in the comparison as tests through GeneXpert were performed only in the four GeneXpert districts. 

#### 2.2.3. Additionality in District Level TB Case Notification Rates

The Nepal NTP reports TB cases on a trimester basis (3 × four-month reporting periods each year). We obtained case notification data disaggregated by age group and gender of all intervention and control districts from the NTP database. 

The contribution of the project to the NTP notification was determined by the additionality method used by TB REACH, which is calculated by subtracting the cases notified during the baseline period from the cases notified during the implementation [15]. The baseline covered the period 16 July 2016 through 15 June 2017, and the implementation period started on 16 July 2017 until 15 Nov 2018.

Second, to calculate the predicted case notification data of the intervention year in both intervention and control districts, we performed a trend analysis using linear regression method for the preceding three year case notification data. We then applied the adjusted additionality method, which calculates the difference between the expected values during the months of implementation compared to the actual notification among bacteriological cases. We compared the trimester data as well as the consolidated data of the intervention period with the estimated case notification data for both intervention and control districts. The changes in number and percentage from the analysed data provided the additionality in the intervention and control districts. We compared the additionality of the intervention and control districts and calculated the total additionality.

Finally, we used the double difference method to evaluate the yield (unadjusted and adjusted) from the strategies.

## 3. Results

### 3.1. Screening and Diagnosis of TB Cases

The cascade of screening to diagnosis of TB cases during the study period is shown in Figure 1. The majority of the individuals diagnosed with TB were male (752/1092; 69%). A fifth of those with TB was aged 65 years and above (230/1092; 21%). (Appendix A).

During the implementation period, among the three interventions, screening through the four hospital OPDs that used GeneXpert for TB diagnosis identified the highest proportion of TB cases (n = 566/1092; 52% cases). Contact tracing in all eight districts yielded a total of 441 cases (40.4% n = 441/1092), and 62 TB camps yielded 85 cases (7.8%; n = 85/1092).

The NNS was lowest for the OPD strategy (NNS = 6), with NNS = 48 for contact tracing, and NNS = 353 for TB camps. The NNT was just 5 for the OPD strategy, 36 for contact tracing, and 94 for TB camps.

### 3.2. Comparison of Yield from GeneXpert vs. Smear Microscopy

Table 1 shows the indicators between the two diagnostic tools implemented in six districts for all the three interventions. The majority of the cases in the study were diagnosed using GeneXpert with yield rate 5.5% vs. 2% by smear microscopy. Using GeneXpert and smear microscopy, respectively, NNS was 38 and 93, and NNT was 18 and 53.

However, as districts using smear microscopy do not have OPD screening as their intervention strategies, we further compared the interventions using contact tracing and TB camps only. The respective yield rate from GeneXpert and smear microscopy during contact tracing was 3.2% and 2.4%, and that for TB camps was 1.4% and 0.82%. The use of GeneXpert for TB diagnosis increased case detection (OR = 1.4; 95%CI = 1.12–1.70; *p* = 0.0026) (Table 1).

### 3.3. Additionality in Case Notification

Figure 2 shows the NTP case notification data for the intervention and control districts during the preceding year (2016/17) and the year of intervention (2017/18 and 2018/19). We calculated additionality and compared the case notification data with the corresponding trimester of the baseline year using unadjusted additionality methods for case notifications for bacteriologically positive TB in the implemented districts (Figure 2). We consolidated the trimester data of intervention district and observed 406 additional cases (20.1%) in bacteriologically confirmed cases. During the same period, there was a decrease in case diagnosis in the control districts of −2.2%. Therefore, the additionality in the intervention districts contributed by the project was 22.3%.

We also compared the case notification with the estimated three year (2014/15–2016/17) trend-adjusted data. We identified additionality of 15.4% for bacteriologically confirmed TB and decrease of 13.9% in the control districts. The comparison showed the total increment in the case notifications in the implemented districts by 29.0% (Figure 3). Since 2014/15, we observed an annual declining linear trend case notification data in the corresponding trimester before intervention. 

Therefore, the results showed that the double difference was substantial in both unadjusted (22.3%) and adjusted additionality (29.0%).

## 4. Discussion

This TB REACH wave 5 project successfully contributed 22% additionality to the NTP case notification in the project districts. Over 50% of the project yield was attributable to screening for TB symptoms in OPD attendees at the government district hospitals and subsequent testing by GeneXpert. Although this strategy is often not considered “active” case finding, in the Nepali context, at the time of the project, this was an extension of the standard National TB programme strategy. An OPD screening strategy known as the FAST strategy has now been incorporated into the Global Fund-supported activities of the NTP in priority districts. The substantial yield of this strategy, when coupled with GeneXpert testing to maximise the sensitivity of the testing, demonstrates that there is an urgent need for every district hospital in Nepal to be equipped with GeneXpert testing and to consistently implement TB symptom screening in OPD attendees.

A further major finding of the project was the substantial yield of cases (n = 230) in individuals >65 years of age. This is consistent with our previous TB REACH wave 2 project data November 2011 to June 2014 and the 2019 prevalence survey and shows that the elderly are among those high-risk groups unable to access existing TB diagnostic services [6,16]. National strategies to improve TB screening and diagnosis among the elderly should be developed and implemented and integrated with existing health services. 

The 2019 prevalence survey in Nepal has revealed a high proportion of asymptomatic individuals with active TB; 70% of those identified were asymptomatic [6]. While our study shows that substantial additionality can be achieved using a simple verbal symptom-screening tool, to achieve the END-TB strategy goals, this must be complemented by a scale-up in availability of chest X-ray diagnosis [17]. Currently, there are rarely specialised radiographers available in remote districts of Nepal, and chest X-ray equipment is often nonfunctioning. The recent development of improved accuracy of computer-assisted chest X-ray reading for TB diagnosis presents an ideal opportunity for Nepal to address this gap [18].

In this study, we obtained a lower yield from TB camps, which were conducted in hard-to-reach areas. Nevertheless, it is important to conduct such camps as the study screened and tested a large number of people who are not able to access diagnosis in other ways. Such strategies improve equity of access to quality TB services, reduce patient incurred costs from TB illness in the most impoverished population groups, and are essential to “leave no-one behind” in countries with large remote rural populations [19,20].

Our results showing the increase in case diagnosis by using GeneXpert is consistent with other studies that highlighted improved case detection through the use of sensitive molecular diagnostic methods [21]. WHO now recommends molecular diagnostic tests such as GeneXpert as the first test of choice to investigate patients for TB; however, resource limitations are a major barrier to uptake in high TB burden or low resource settings. To achieve widespread scale-up of GeneXpert as a first line test for TB through national policies, cost-effectiveness studies are necessary to quantify the relative costs of health systems approaches applying GeneXpert or smear. We are currently preparing a health system costing evaluation for publication. These studies will inform NTP to design and implement effective and sustainable case-finding strategies [22,23].

To achieve the END TB strategy goals and close the global case detection gap, it is essential that all presumptive TB cases in high burden settings are tested by rapid molecular diagnostic methods. During the implementation of the project, Nepal underwent a major change in its political structure to a federal system, which included restructuring of the public health system administration. This affected the regular functioning of health workers, including those working in TB, and could have influenced the case notification in both intervention and control districts. The short duration of the project period exacerbates the weight of such external challenges. Therefore, to implement the ACF strategies, a significant period should be allocated prior to the project inception, especially for establishing human resources and rapport building within the health system to ensure optimal integration and synergy.

## 5. Conclusions

Substantial additionality in TB case notification was demonstrated through OPD screening and social-contact-tracing strategies. Higher yields were achieved using GeneXpert than smear microscopy for active case finding. Although TB camps had a relatively low yield, this strategy reaches remote populations and is an important component of a comprehensive TB case-finding strategy. Comprehensive cost-effectiveness studies that evaluate the monetary value of these interventions would better contribute evidence for the design and application of optimal NTP strategies to achieve the END TB targets [24,25].

## Figures and Tables

**Figure 1 tropicalmed-06-00050-f001:**
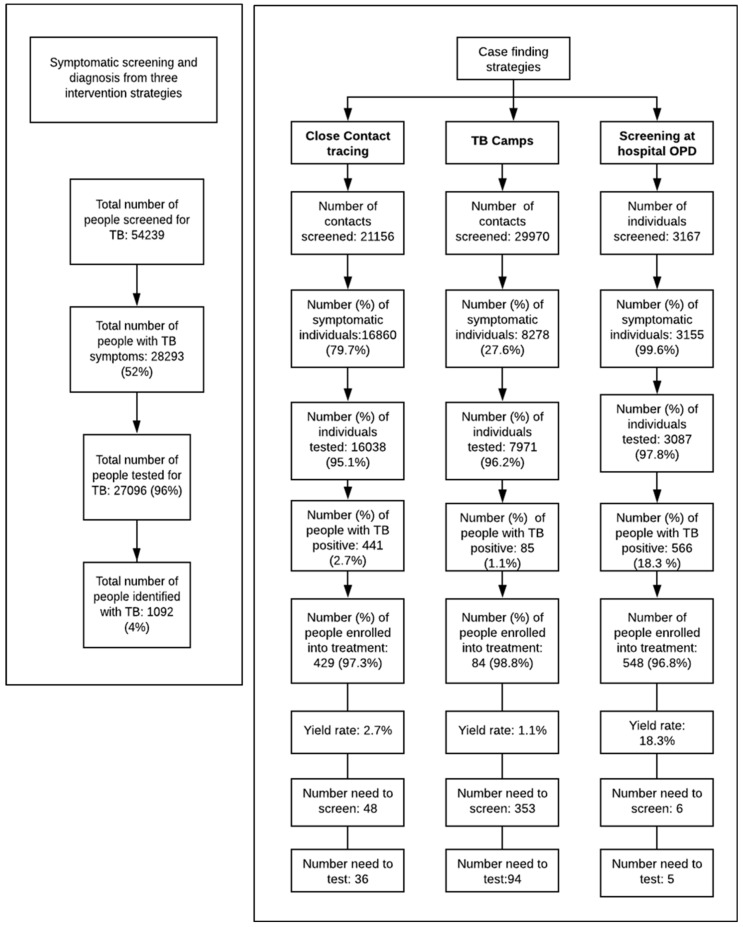
Flowchart showing the screening to diagnosis and treatment process in the intervention district.

**Figure 2 tropicalmed-06-00050-f002:**
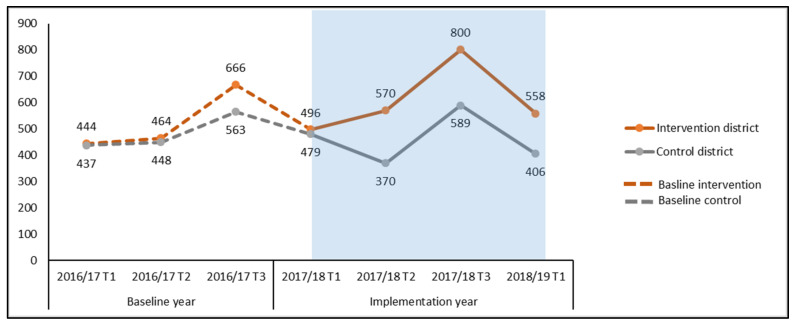
Comparison of case notification before and after implementation of active case finding (ACF) interventions (unadjusted data).

**Figure 3 tropicalmed-06-00050-f003:**
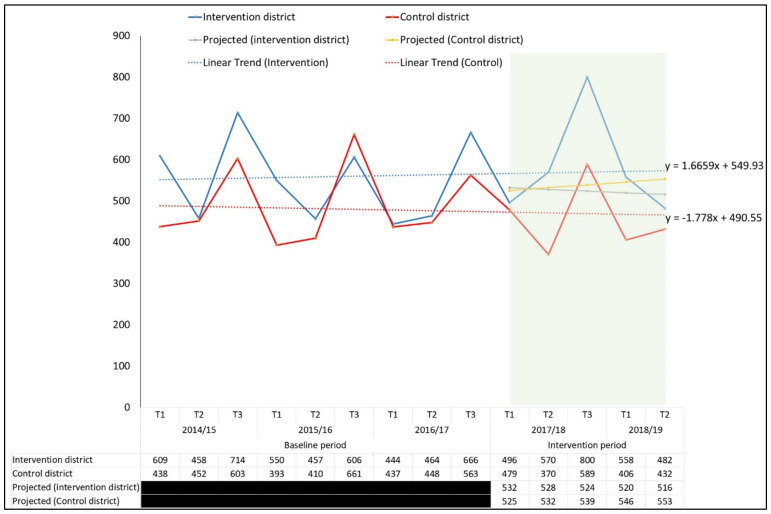
Comparison of case notifications before and after implementation of ACF interventions (3 year trend-adjusted, i.e., estimated case notification).

**Table 1 tropicalmed-06-00050-t001:** Comparison between districts using GeneXpert and smear microscopy as (tuberculosis) TB diagnostic tool (excluding Salyan and Arghakhanchi districts).

Indicators		Districts Diagnosing TB with GeneXpert	Districts Diagnosing TB with Smear Microscopy
Total number of people screened		32,616	11,202
Total number of people with TB symptoms		16,060	6,575
Total number of people tested for TB		15,637	6,309
Total number of people diagnosed with TB positive		859	120
Yield rate (%)	Contact Tracing	3.2 (273/8567)	2.4 (103/4246)
	TB Camps	1.4 (58/4106)	0.82 (17/2063)
	Three interventions combined	5.5 (859/15,637)	2 (120/6309)
Numbers needed to screen (NNS)		38	93
Numbers needed to test (NNT)		18	53

## Data Availability

The datasets generated and analysed during the study are available upon request to the corresponding author.

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
