# Peer review of "Comparative Yield of Tuberculosis during Active Case Finding Using GeneXpert or Smear Microscopy for Diagnostic Testing in Nepal: A Cross-Sectional Study"

_tropicalmed, 2021, doi:10.3390/tropicalmed6020050_

Round 1

Reviewer 1 Report

The manuscript by Gurung et al is a good document comparing the positivity between smear microscopy and GeneXpert in the Nepalese setup and will be ideal source material for future work in the field. The research work has enabled the team to find more cases but also identify the direction of future policy decision. The findings are not novel, but they are unique in suggesting GeneXpert can be used in the outpatient department in countries with limited resources.

Author Response

Response to Reviewer 1

 Comment 1: The manuscript by Gurung et al is a good document comparing the positivity between smear microscopy and GeneXpert in the Nepalese setup and will be ideal source material for future work in the field. The research work has enabled the team to find more cases but also identify the direction of future policy decision. The findings are not novel, but they are unique in suggesting GeneXpert can be used in the outpatient department in countries with limited resources.

Response 1: We thank the reviewer for these positive comments, recognizing the significance of our findings to influence policy and implementation of GeneXpert scale-up in LMIC settings. We agree that global studies have shown the GeneXpert machines have increased sensitivity compared to smear and aimed to quantify the increased yields achieved when implementing Xpert testing instead of smear microscopy for active TB case finding in LMIC in order to provide evidence to policymakers. We agree that the finding of high TB case yields when GeneXpert is applied in out-patient department screening is particularly important to change practice. This is an important recommendation for the National Tuberculosis Program in Nepal and other similar settings to implement evidence-based case finding strategies and increase the case diagnosis to achieve the END TB target.

Reviewer 2 Report

The study describes the yield from active case finding in Nepal comparing GeneXpert and smear microscopy as diagnostic methods. In general, the study is impressive in its size and coverage, and the design and results are fairly clear and digestible. However, there are a number of typos detected and the in-text number reference looks a bit odd (consistently placed after the period at the end of the sentence) and should be double checked.

The following are some specific comments:

Ln 113: What is the manner of sputum collection? morning? voluntary/induced?

Ln 148: No information on GeneXpert and smear microscopy method provided; albeit smear microscopy is a standard diagnostic method there are some variations that alter the sensitivity. 

Ln 157: This subsection title is vague. Perhaps retitle it to ‘Predicted case notification rate’ or similar

Ln 178: This statement should come much earlier in the first few paragraphs of the subsection.

Ln 285 - 287: This is an important point to address in the discussion. Without cost-effectiveness data it is difficult to conclude that the increased yield and reduced NNS/NNT from using GeneXpert is the way to go to reach the end TB goal.

Author Response

Response to Reviewer 2

Comment 1: The study describes the yield from active case finding in Nepal comparing GeneXpert and smear microscopy as diagnostic methods. In general, the study is impressive in its size and coverage, and the design and results are fairly clear and digestible. However, there are a number of typos detected and the in-text number reference looks a bit odd (consistently placed after the period at the end of the sentence) and should be double checked.

Response 1: We thank the reviewer for these comments on the importance and scope of the study. We apologise for the typos in the manuscript and have carefully re-proofed the manuscript to correct all typos throughout. We have also edited the in-text referencing style to place the citation within the period, as the reviewer suggested, throughout the manuscript.

Comment 2: Ln 113: What is the manner of sputum collection? morning? voluntary/induced?

Response 2: The process of sputum collection is described in lines 112-115. No sputum induction was conducted. We have made a clarification with regard to the sputum collection method:

“The CHVs collected one morning voluntary sputum sample for GeneXpert testing or two samples (spot and morning) for smear microscopy. No sputum induction techniques were used but participants were given instructions by CHVs on how to produce quality sputum. On the day of sample collection, the CHVs transported the samples to the nearest microscopy or GeneXpert testing center.” (Ln 112- Ln 115)

Comment 3: Ln 148: No information on GeneXpert and smear microscopy method provided; albeit smear microscopy is a standard diagnostic method there are some variations that alter the sensitivity. 

Response 3: We agree with the reviewer that variation in smear microscopy protocols can influence sensitivity of the test, which is one of the major weaknesses of smear microscopy. The standard WHO methodology was applied in this case, following WHO NTP guidelines applied throughout Nepal. GeneXpert testing followed the standard protocols recommended by Cepheid, without variation. To avoid long protocol descriptions of standardized methodology in the text, for clarity, we have added the following statement and references:

Standard Operating Procedures for smear microscopy and GeneXpert testing followed the standard Nepal NTP guidelines, which are based on the relevant WHO protocols and the manufacturer’s standard operating procedure [11,12]. (Ln 97- Ln 100).

Comment 4: Ln 157: This subsection title is vague. Perhaps retitle it to ‘Predicted case notification rate’ or similar

Response 4: We have revised this title to ‘Additionality in district level TB case notification rates.’ (Ln 169) The section does not describe predicted case notification rates, but actual case notification rates achieved.

Comment 5: Ln 178: This statement should come much earlier in the first few paragraphs of the subsection.

Response 5: We thanks the reviewer for this suggestion. We agree this improves clarity for the reader and have moved the text as suggested. We have updated the statement under ethical approval in the line number Ln 148- Ln 151

Comment 6: Ln 285 - 287: This is an important point to address in the discussion. Without cost-effectiveness data it is difficult to conclude that the increased yield and reduced NNS/NNT from using GeneXpert is the way to go to reach the end TB goal.

Response 6: We agree with the reviewer that this issue is of critical importance and have expanded the discussion of this point in the text in lines Ln 282-Ln 289:

WHO now recommends molecular diagnostic tests such as GeneXeprt as the first test of choice to investigate patients for TB, however resource limitations are a major barrier to uptake in high TB burden or low resource settings. To achieve widespread scale-up of GeneXeprt as a first line test for TB through national policies cost effectiveness studies are necessary to quantify the relative costs of health systems approaches applying GeneXpert or smear. We are currently preparing a health system costing evaluation for publication. These studies will inform NTP to design and implement effective and sustainable case finding strategies [22,23].

Reviewer 3 Report

Please add % in the flow chart 

Please provide demographics of the people screened

Author Response

Response to Reviewer 3

Comment 1: Please add % in the flow chart 

Response 1: We thank the reviewer for this comment to improve clarity of the information presented in the figure. suggestion. We have now added the percentages to the flow chart (Ln 200- Ln 201).

Comment 2: Please provide demographics of the people screened

Response 2: We agree with the reviewer that this information would add to the interpretation of the data. However, the limited budget for research did not allow us to collect this data in the field at the time of implementation and therefore we are unable to add it to the analysis at this stage.

Round 2

Reviewer 2 Report

After addressing some of the questions I think this manuscript is good to go.